# The Rise of Fine-Tuned CAR-Based Therapies Against Acute Myeloid Leukemia

**DOI:** 10.3390/cancers17243892

**Published:** 2025-12-05

**Authors:** Alejandro Segura Tudela, Ron Geller, Bruno Paiva, Sara Carmen Torres Sánchez, Elisa González Romero, Pilar Lloret Madrid, Pedro Chorão, Javier de la Rubia, Pau Montesinos, Manuel Guerreiro

**Affiliations:** 1Instituto de Investigación Sanitaria La Fe (IISLAFE), 46026 Valencia, Spain; alejandro_segura@iislafe.es (A.S.T.); sara_torres@iislafe.es (S.C.T.S.); elisa_gonzalez@iislafe.es (E.G.R.); pilar_lloret@iislafe.es (P.L.M.); pedro_chorao@iislafe.es (P.C.); delarubia_jav@gva.es (J.d.l.R.); 2Institute for Integrative Systems Biology (I2SysBio), Universitat de València-CSIC, C. Catedràtic José Beltrán 2, 46980 Paterna, Spain; ron.geller@csic.es; 3Clinica Universidad de Navarra (CUN), Centro de Investigacion Medica Aplicada (CIMA), Instituto de Investigacion Sanitaria de Navarra (IDISNA), CIBER-ONC Number CB16/12/00369, 31008 Pamplona, Spain; bpaiva@unav.es; 4Hematology Department, Hospital Universitari i Politècnic La Fe, 46026 Valencia, Spain; montesinos_pau@gva.es; 5School of Medicine and Dentistry, Catholic University of Valencia, 46001 València, Spain; 6Department of Medicine, University of Valencia, 46010 Valencia, Spain

**Keywords:** acute myeloid leukemia, CAR T cells, next-generation CAR therapies, AML antigen landscape, logic-gated CAR circuits, armored CAR T cells, gene-edited immune cells, CAR-NK cells, CAR-γδ T cells, allogeneic CAR products

## Abstract

Chimeric antigen receptor (CAR) therapies have changed the treatment paradigm for several hematologic malignancies. However, their success in acute myeloid leukemia (AML) has been limited due to associated side effects and the complexity of the disease. In recent years, next-generation CAR therapies have been developed to overcome these challenges. These innovative approaches include modulating antigen-binding affinity, combining multiple targets into logical circuits, using switchable or modular receptors that can be turned on or off, incorporating proteins that enhance antitumor activity, applying gene-editing tools, and exploring alternative immune cells such as natural killer cells or macrophages. This review examines these emerging technologies and their potential to guide the development of safer and more effective engineered cell therapies for AML.

## 1. Introduction

Chimeric antigen receptors (CARs) are synthetic, modular proteins that redirect immune cell specificity toward previously predefined antigens. Their design integrates elements of adaptive and synthetic immunology, enabling immune cells, commonly T lymphocytes, to acquire de novo targeting capabilities against disease-associated antigens, bypassing the requirement of major histocompatibility complex (MHC)-restricted recognition [1,2,3,4].

This design enables CAR T cells to recognize and eliminate tumor cells independently of peptide-MHC, achieving remarkable clinical success against B-lineage malignancies, including B-acute lymphoblastic leukemia, diffuse large B-cell lymphoma, and multiple myeloma, and resulting in several clinically approved products [1,2,3,4].

However, this prototypical structure is not absent of significant limitations. Relapse driven by antigen escape, T-cell exhaustion, limited persistence, and severe toxicities, such as cytokine release syndrome (CRS) and immune effector cell-associated neurotoxicity syndrome (ICANS), remain as major challenges, even in diseases where CAR T therapy is already approved. These CAR-associated challenges have driven the development of next-generation CAR architectures, incorporating protein-secreting domains, switchable activation modules, multi-antigen targeting strategies, and gene-editing approaches to enhance both efficacy and safety, many of which have been primarily investigated in the context of solid tumors [1,5].

In acute myeloid leukemia (AML), the most common acute leukemia in adults, CAR T cell therapies face additional obstacles. The lack of AML-specific antigens represents a major barrier, as most targets (e.g., CD33, CD123, CLEC12A) are also expressed on normal hematopoietic stem and progenitor cells (HSPCs), creating a risk of on-target, off-tumor myelotoxicity and prolonged marrow aplasia. Moreover, the immunosuppressive bone marrow microenvironment, characterized by inhibitory cytokines, regulatory myeloid cells, and metabolic constraints, limits CAR T cell expansion, persistence, and effector function [4,6,7].

In addition, specific molecular and clinical subsets of AML represent particularly high-risk contexts. MECOM-driven AML and AML transformed from myeloproliferative neoplasms (post-MPN AML) frequently exhibit aggressive clonal evolution, profound chemoresistance, and extensive remodeling of the bone marrow microenvironment. TP53-mutated AML displays marked baseline T cell dysfunction and senescence, and recent single-cell analyses have shown that ASXL1- and RUNX1-mutated AML share this phenotype, characterized by a predominance of senescent-like CD8^+^ T cells [8,9,10,11,12,13]. Collectively, these alterations reduce the pool of functional lymphocytes available and may hider both the manufacturing process and the in vivo potency of autologous CAR T cell products.

These challenges underscore the urgent need for innovative CAR engineering strategies, including multi-antigen recognition to mitigate escape, gene editing to enhance potency or safety, regulatable formats to control activity in vivo, and alternative immune cell platforms (e.g., NK, NKT, γδ T cells), to expand the applicability of CAR-based therapies in AML. In this review, we summarize recent advances in CAR design and discuss their potential to overcome disease-specific obstacles and improve clinical translation for AML.

## 2. Rationale, Scope, and Structure of the Review

To delineate the scope of this review, we first outline the canonical CAR architecture and summarize the thematic organization of the manuscript.

Canonical CAR architecture is composed of three principal modules (Figure 1): (i) an extracellular antigen-binding domain, typically a single-chain variable fragment (scFv) derived from an antigen-specific monoclonal antibody; (ii) a hinge and transmembrane domain, which confer structural flexibility and stably anchor the receptor to the plasma membrane; and iii) an intracellular signaling endodomain, usually consisting of one or more costimulatory modules (CD28 or 4-1BB) linked to the CD3ζ activation motif.

In the following sections, we explore a range of emerging engineering strategies developed to overcome the biological and clinical challenges of AML. These include:Affinity modulation to fine-tune antigen recognition and reduce toxicity.Logic-gated CARs (OR, AND, NOT, IF-BETTER, IF-THEN) to enhance selectivity and mitigate antigen escape.Modulable CAR platforms, including adapter-based systems and ON/OFF or suicide switches for dynamic control.Armored CARs, equipped with cytokine secretion, membrane-bound proteins, or immune engagers to boost potency.Gene-edited CARs, leveraging disruption or overexpression of key genes to improve function and scalability.Alternative immune cell platforms, such as CAR-NK, CAR-γδ T, and CAR-macrophages, to bypass limitations of αβ T cells.

Together, these approaches represent a multifaceted effort to refine CAR-based immunotherapy for AML and accelerate its clinical translation.

The content of this review is based on a curated selection of publications. Relevant studies were identified through searches in PubMed, Web of Science, Scopus, and ClinicalTrials.gov using combinations of terms such as ‘AML’, ‘acute myeloid leukemia’, ‘CAR T’, ‘CAR NK’, ‘cell therapy’, ‘CLL1’, ‘CD123’, ‘dual CAR’, ‘affinity tuning’, ‘adapter CAR’, ‘allogeneic CAR’, ‘γδ T’, ‘NKT’, ‘safety switch’, and related keywords. We included original preclinical and clinical studies, ongoing trial reports, and high-impact reviews with direct relevance to CAR engineering strategies. Eligible publications comprised work on AML-directed CAR T, CAR NK, CAR NKT, γδ T, or modular/adaptable platforms, as well as studies detailing mechanistic aspects of antigen selection, CAR architecture, affinity modulation, gene editing, safety switches, or multi-antigen targeting. Clinical trial data, both published and registry-listed, were also incorporated. Non-peer-reviewed content and abstracts lacking sufficient methodological detail were excluded. Reference lists of key articles were manually screened to identify additional relevant work, and all selected records were curated using Mendeley to organize studies by antigen target, engineering approach, and stage of clinical development.

## 3. Modulation of CAR Affinity

While initial CAR designs were based on clinically approved high-affinity antibodies (e.g., Cetuximab and Trastuzumab), recent evidence has indicated that the modulation of CAR expression and ligand-affinity can impact both treatment efficacy and safety [14,15]. An analysis of 38 CART clinical trials for solid tumors reported increased efficacy and reduced toxicity for moderate-affinity CAR T cells compared to high-affinity CART [14]. However, these data are derived almost entirely from solid tumor antigens, particularly CLDN18.2 and GD2, and the studies do not control for key confounders such as antigen density, epitope accessibility, or CAR tonic signaling, making it difficult to extrapolate these findings to AML.

Modulation of CAR affinities has been implemented using rational structural-based approaches, phage-display, and various mutagenesis methods, including alanine scanning, error-prone PCR, or saturation mutagenesis techniques. However, the selection of optimal CART on and off rates as well as expression levels for achieving maximal efficacy and reduced toxicity remain to be defined. In the context of AML, CARs with lower affinity to two potential ligands of relevance to AML, CD123 [16] and CD38 [17], have been designed and shown to have improved outcomes compared to their high-affinity counterparts; nevertheless, available data remain sparse and insufficient to establish affinity thresholds suitable for clinical translation. Affinity-tuned CARs may hold particular relevance in AML contexts characterized by low antigen density, such as relapsed/refractory (R/R) disease, where antigen downregulation contributes to immune escape.

## 4. Logic-Gated CAR T Cells

AML antigen heterogeneity is associated with antigen escape, with the subsequent development of disease relapse, or unacceptable on-target off-tumour toxicities, both considered major limitations of CAR T cell therapy. To address these issues, multispecific next-generation CAR T cells employ Boolean logic-based targeting strategies (Figure 2). Accordingly, a diverse and expanding set of logic-gated designs, including OR, NOT, AND, IF-BETTER, and IF-THEN-gated CAR T cells, are being tested in AML to improve both efficacy and safety [18].

### 4.1. OR-Gate

Multispecific OR-gated targeting can be achieved through several approaches: (i) co-infusion of two different CAR T cell products, (ii) producing dual CAR approaches through transduction with bicistronic vectors or co-transduction with multiple vectors, or (iii) using a bivalent CAR construct (tandem or loop CAR) [1,4,18]. The principle behind these strategies is that recognition of either of the two or more target antigens is sufficient to trigger tumor cell killing (Figure 2a).

OR-gated CAR T cell approaches have mainly been evaluated in preclinical studies. Dual CAR T cells have been generated with bicistronic vectors targeting combinations such as CD123/NKG2DL, CD123/CLEC12A, and CD123/CD33 [19,20,21], as well as by co-transduction strategies for CD123/CLEC12A [22]. In addition, several bivalent CAR constructs have been tested in tandem configurations (e.g., CD123/GRP78, CD123/CLEC12A, CD123/FRβ) [23,24,25,26] and in loop formats (e.g., CD123/CD33) [27]. Dual CAR T cells using bicistronic vectors against CD33/CLEC12A and CD123/CLEC12A are currently under evaluation in phase I and phase II/III clinical trials, respectively (NCT05016063, NCT03795779, NCT05654779, NCT03631576).

Beyond dual specificity, trispecific OR-gated CARs have been explored in B-cell malignancies [28,29], with some constructs now being tested in clinical trials (NCT05418088). Recently, a trivalent OR-gated CAR, known as the extracellularly linked concatemeric trivalent cytokine (ELECTRIC) CAR, was evaluated in preclinical AML models. This second-generation CAR targets KIT, MPL, and FLT3 via a tandem fusion of their respective ligands (SCF, TPO, and FLT3LG) and incorporates CXCR4 overexpression to enhance migration toward bone marrow niches [30]. OR-gated targeting thus broadens the repertoire of addressable antigens and may reduce the risk of antigen escape. However, it also entails the need for careful selection of antigen pairs with minimal risk of cumulative on-target toxicity to healthy tissues. In addition, although antigen escape is clearly demonstrated in preclinical AML models [4,6,31], clinical evidence of antigen-loss relapse after CAR T cell therapy remains limited. Most AML trials have not achieved sufficiently deep or durable responses for antigen-negative outgrowth to become detectable. Thus, OR-gated designs address a biologically plausible and experimentally supported mechanism, although definitive clinical evidence is still lacking.

### 4.2. AND-Gate

In contrast to OR logic, which prioritizes inclusivity of targets, AND-gated strategies aim for higher stringency, activating CAR T cells only when two antigens are co-expressed (Figure 2b). AND-logic circuits were designed to avoid, or at least reduce, the on-target/off-tumour toxicities. These constructs typically involve the co-expression of two complementary CARs: one harboring the CD3ζ activation domain and the other functioning as a chimeric co-stimulatory receptor (CCR) containing only a signaling motif such as CD28 or 4-1BB. Each CAR recognizes a distinct antigen, and full T cell activation is achieved only when both antigens are simultaneously engaged [1,4,5]. An alternative AND-gate strategy replaces CD3ζ or co-stimulatory domains with proximal TCR signaling adaptors, such as LAT or SLP76, with the advantage of enforcing stricter AND-gating, since each CAR delivers only a partial signal, thereby reducing off-tumor activation and preserving more physiological TCR signaling dynamics, which may improve both safety and efficacy [4,18,32,33].

This dual-antigen recognition strategy should enable selective elimination of tumor cells co-expressing both target antigens while sparing healthy tissues that express only one target antigen. Despite its promise, AND-gate CAR designs have been evaluated in AML only once, using a low-affinity mutant IL-3–CD3ζ/CD33-CCR construct [34]. Effective implementation in AML will require identification of antigen pairs that are consistently co-expressed on leukemic blasts but absent from critical healthy hematopoietic and non-hematopoietic cells. A major limitation of this approach is its reliance on stable antigen co-expression, as loss or downregulation of either target could permit immune evasion and drive disease relapse.

### 4.3. NOT-Gate

While AND logic enhances stringency, it may limit efficacy against heterogeneous tumors. This has motivated the development of complementary strategies to mitigate on-target/off-tumor toxicities using NOT-gated CAR T cell designs. In this approach, a conventional activating CAR specific for antigen A is co-expressed with an inhibitory CAR (iCAR) directed against antigen B. The iCAR incorporates an intracellular inhibitory signaling domain, typically containing immunoreceptor tyrosine-based inhibitory motifs (ITIMs). Cytotoxicity is triggered only in cells expressing antigen A in the absence of antigen B (“A NOT B”), whereas cells co-expressing both antigens are spared due to the dominant inhibitory signal delivered by the iCAR (Figure 2c). In this manner, antigen B functions as a protective marker, shielding healthy cells from CAR T cell-mediated cytotoxicity [4,5,18,33].

This therapeutic concept has been investigated in AML exclusively in preclinical models, using combinations such as CD93/CD19-iCAR, CD33/HLA-DR-iCAR, CD15-iCAR/CLEC12A, and CD16-iCAR/CLEC12A [35,36,37]. Furthermore, NOT-gate strategies have been integrated with OR-gated designs to further refine specificity, as demonstrated by constructs such as CD33 OR FLT3/EMCN-iCAR and CD33 OR SPN/CD16b-iCAR OR CLEC9A-iCAR [38,39].

### 4.4. IF-BETTER-Gate

Whereas NOT-gates impose a “brake” on activation, IF-BETTER-gates instead function as a “booster,” enhancing CAR sensitivity when a helper antigen is present. Strategies based on IF-BETTER-gated targeting rely on the co-expression of a conventional second-generation CAR directed against antigen A and a secondary “helper” CCR targeting antigen B. In this configuration, CAR sensitivity to antigen A is significantly enhanced when antigen B is present (“IF B, BETTER A”), enabling a context-dependent activation mechanism (Figure 2d). This context-dependent scenario allows for selective targeting of tumor cells co-expressing both antigens, while limiting cytotoxicity against normal tissues that express low levels of antigen A in the absence of B [18,40].

The therapeutic potential of this approach has been demonstrated in preclinical models using T cells co-expressing a calibrated ADGRE2-CD28-CD3ζ1XX CAR and a CLEC12A-41BB CCR (ADCLEC.syn1). These modified T cells showed reduced toxicity and improved efficacy against antigen-low AML variants [41], and the construct is now being evaluated in a phase I clinical trial (NCT05748197).

The IF-BETTER logic circuit represents a promising approach to improve CAR T cell sensitivity against AML variants exhibiting low levels of target antigen expression while limiting off-tumor effects. However, as in the case of AND logic circuits, this requires the identification of two antigens that are consistently co-expressed on leukemic cells but not healthy tissues.

### 4.5. IF-THEN-Gate

Unlike IF-BETTER, which modulates CAR sensitivity quantitatively, IF-THEN-gates introduce a sequential, conditional activation mechanism. Synthetic Notch (synNotch) receptors enable spatially and temporally controlled activation. Recognition of a first antigen by the synNotch receptor triggers release of a transcription factor, which in turn induces expression of a CAR directed against a second antigen, thereby linking recognition of both antigens through a gene circuit (Figure 2e) [4,18,42]. This two-step mechanism allows not only cell-specific recognition but site-specific T cell activation, enhancing tumor selectivity and reducing off-target toxicity [43].

This approach was first validated by Roybal et al. [44]. Since its initial development, the application of the synNotch platform has been progressively expanded, with a predominant focus on solid tumors. Its implementation in the AML context was demonstrated in a CD33-SynNotch→CD12 logic circuit [45].

While the synNotch platform offers substantial versatility, it also presents limitations, including ligand-independent activation and the relatively short half-life of CAR expression following synNotch disengagement [42]. These challenges, along with the difficulty of selecting optimal antigen pairs, and issue that is also shared by AND and IF-BETTER logic circuits, underscore the need for further refinement.

## 5. Modulable CAR Specificity

Beyond gated designs, an alternative strategy to increase flexibility relies on modular CAR platforms, which decouple antigen recognition from signaling. Modular CAR platforms have been designed to confer adaptable antigen specificity through an externally controllable, two-component system. These systems generally consist of: (i) a membrane-bound CAR-like receptor, and (ii) a soluble adapter molecule that recognizes a tumor-specific/associated antigen. Importantly, both components are engineered with mutual affinity, ensuring that functional CAR signaling occurs only when the adapter simultaneously engages its target antigen and the CAR receptor (Figure 3a). This configuration enables precise and dynamic control of CAR T cell activity in real time [4,5,46,47].

Several modular CAR architectures have been developed and evaluated in preclinical and clinical AML settings:Universal CAR (UniCAR): the CAR recognizes a defined tag epitope present on the adapter [48,49,50] (NCT04230265).Reversed CAR (RevCAR): the adapter recognizes a specific tag on the CAR extracellular domain [51].Split, Universal, and Programmable (SUPRA) CAR: based on leucine zipper domains, comprising a membrane-bound CAR endodomain core with a leucine zipper extracellular domain, and a soluble tumor-binding domain fused to a complementary leucine zipper [52]. This last approach has not yet been investigated in AML.

Modular CAR platforms provide a promising path toward personalized AML treatment, particularly in light of its heterogeneous antigenic landscape. This strategy allows individualized target selection by choosing the most appropriate tumor-specific adapters from pre-established libraries. In addition, the modular design permits rapid therapeutic adaptation: as tumors evolve or escape immune pressure, the adapter can be exchanged accordingly, thereby sustaining antitumor efficacy while minimizing toxicity.

Despite being a highly versatile platform, modular CAR approaches have demonstrated reduced T cell efficacy compared to conventional CARs, often requiring higher CAR T cell doses in preclinical models to achieve similar antitumor effects.

## 6. CAR Formats with Modifiable or Limited Activity

A major challenge in CAR T cell therapy remains the management of severe toxicities, particularly cytokine release syndrome (CRS) and immune effector cell-associated neurotoxicity syndrome (ICANS). To address these complications, next-generation CAR designs incorporate regulatable systems that allow temporal control or conditional elimination of CAR T cells in response to external drugs [1,3,5].

### 6.1. ON/OFF Switches

ON/OFF switch systems allow reversible modulation of CAR activity (Figure 3b). Pharmacologic inhibition of CAR signaling with tyrosine kinase inhibitors, such as dasatinib, can transiently suppress CAR function, effectively serving as an inducible “pause button” [53]. Alternative approaches embed an off-switch directly into the CAR construct. For instance, the SWIFF-CAR platform integrates a self-cleaving degradation motif regulated by a protease/protease inhibitor pair, allowing reversible control of CAR surface expression through the small molecule asunaprevir [54]. Another strategy involves drug-induced dimerization of split CAR designs, where two receptor fragments assemble into a functional CAR only in the presence of a small molecule; this approach is currently being evaluated clinically in AML [55] (NCT05105152).

### 6.2. Suicide Switches

By contrast, suicide switches provide an irreversible safety mechanism to eliminate CAR T cells when toxicities are unmanageable (Figure 3c). One approach is the incorporation of truncated surface markers such as EGFRt, CD20 epitopes, or HER2t, enabling selective depletion of CAR T cells with clinically approved monoclonal antibodies [56,57,58,59]. This strategy has been widely applied in CAR T cells targeting AML antigens [60] (NCT03190278; NCT04318678; NCT0392726; NCT02159495). A commonly used platform is the inducible caspase-9 (iC9) system, which triggers rapid, drug-controlled apoptosis to provide an emergency shut-off mechanism [61,62,63,64].

Collectively, these designs expand the safety toolbox for CAR T cell therapies, offering both reversible activity control and irreversible elimination options. Increasingly, such features are being incorporated into AML-targeted CAR platforms to improve clinical safety profiles.

## 7. Armored CAR T Cell

While switchable platforms enhance safety, another major innovation has focused on “arming” CAR T cells with additional functions to improve their potency and overcome the immunosuppressive tumor microenvironment. Fourth-generation CAR T cells, commonly known as “armored” CAR T cells, are a subset of conventional CAR T cells engineered to express additional proteins, either secreted or membrane-bound, that act as extra weapons to fight and eliminate tumors more effectively. These modifications are intended to enhance cytotoxicity, proliferation, persistence, and tumor infiltration, as well as to modulate the tumor microenvironment or recruit bystander immune cells [4,65,66,67].

### 7.1. TRUCK CAR T Cell

One major armoring strategy involves TRUCKs (T cells Redirected for Universal Cytokine-mediated Killing), which are CAR T cells engineered to secrete cytokines in response to activation (Figure 4a). This approach has been extensively studied as a way to reshape the tumor microenvironment (TME) [4,31,65,66,67].

In AML, preclinical studies have shown that IL-15-secreting CAR T or NK cells targeting CLEC12A or CD123 display enhanced cytotoxicity, resistance to exhaustion, and improved survival in xenograft models. However, persistent IL-15 signaling has been associated with severe toxicities, including CRS [68,69,70]. In addition, a TRUCK CAR T cell against CLEC12A engineered to secrete IL-18 is currently under evaluation in a phase I clinical trial for AML (NCT06017258).

### 7.2. Membrane-Bound Protein Modulating CAR T Cell

Another way to “equip CAR T cells with weapons” is through expression of membrane-bound proteins (Figure 4b). Among these, membrane-bound cytokines such as membrane-bound IL-15 have been explored to improve persistence, tumor infiltration, and cytotoxicity. Clinically, this has been implemented in T cells co-expressing membrane-bound IL-15 and a TRuC (T-cell receptor fusion construct) targeting CD70 [71] and the UltraCAR T^TM^ platform, which integrates a membrane-bound IL-15 with a CD33-directed CAR T cell. The latter being evaluated in a phase I/Ib clinical trial (NCT03927261) [72].

Another membrane-bound protein to complement CAR are chemokine-receptors, the overexpression of chemokine receptors has been explored to improve homing of CAR T cells to tumor niches [73]. This approach has been explored in anti-CD25 CAR T cells [74], and notably in the ELECTRIC-CAR platform [30], via overexpression of CXCR4 to enhance migration toward bone marrow.

Lastly, membrane-bound proteins can also be engineered to be reactive or insensitive against inhibitory signaling pathways or even prevent adverse events such as CRS [75]. This includes cytokines coupled to their own receptors such as constitutively active receptors; switch receptor chimeras, which convert inhibitory signals into activating ones by fusing inhibitory extracellular domains with activating intracellular domains; and dominant-negative receptors, which block suppressive pathways by subtraction of the inhibitory intracellular domains from inhibitory receptors [76]. For example, an anti-CD70 CAR γδ-T cells armored with dominant-negative TGFβRII has been tested in preclinical models against different hematological tumor lineages [77].

### 7.3. Antibody-Secreting CAR T Cell

In addition to cytokines and receptors, CAR T cells can also be armed with antibody-based payloads, enabling recruitment of bystander immune effector cells. CAR T cells can be engineered to secrete antibodies or antibody-like molecules, particularly immune engagers, to enhance tumor killing (Figure 4c). These include bi- or multi-specific T-cell engagers (BiTEs, TriTEs) and NK-cell engagers (BiKEs, TriKEs, TetraKEs), which recruit endogenous immune cells to the tumor site [78,79,80].

A major limitation of recombinant immune engagers is their short in vivo half-life, but this can be overcome by engineering CAR T cells to continuously secrete them [1,33]. One early example was a CAR T cell targeting glioblastoma that secreted a T-cell engager to prevent antigen escape [81]. This strategy has since been extended to arm both CAR T and non-CAR-engineered T cells for the treatment of solid tumors and B-cell malignancies [82,83,84].

In AML, two preclinical CAR T models, targeting CD70 and IL10R, respectively, have incorporated the secretion of a CD33/CD3 BiTE. These dual-function constructs successfully recruited bystander T cells and reduced leukemic burden in preclinical models [85,86].

Although alternative strategies to armor CAR T cells have been explored in solid tumors, including the secretion of enzymes such as heparanase [87], the overexpression of intracellular enzymes as catalases [88], the delivery of soluble antitumor proteins [89], or the expression of intracellular enzyme-blocking proteins to enhance CAR T cell activation and persistence [90], these approaches have not yet been investigated in the context of AML.

## 8. Gene-Modified CAR T Cell

Beyond modifications of the canonical CAR design and the incorporation of synergistic functional modules, precise gene editing has emerged as a powerful strategy to enhance the efficacy, safety, and scalability of CAR T cell therapy (Figure 5). Early genome engineering platforms such as zinc finger nucleases (ZFNs) and transcription activator-like effector nucleases (TALENs) enabled targeted disruption of selected genes. However, the advent of CRISPR/Cas systems has revolutionized gene editing by shifting from protein-based to programmable, RNA-guided platforms. These technologies have been applied to generate allogeneic (“off-the-shelf”) CAR T cells, prevent fratricide, disrupt checkpoint inhibitory pathways, and enhance CAR T cell persistence and function [91,92,93].

In addition to gene disruption strategies, the overexpression of key components of antigen receptor signaling pathways has emerged as a complementary approach to potentiate CAR T cell function. Enhancing the expression of intracellular signaling molecules or co-stimulatory adapters can amplify T-cell activation upon antigen engagement [1,6,33].

### 8.1. Disruption to Enhance Function

Gene disruption represents a powerful approach to optimize CAR T cell performance by removing intrinsic barriers that limit activation, persistence, or cytotoxicity (Figure 5a). For instance, silencing inhibitory checkpoint molecule, such as PD-1 (PDCD1) in CAR T cells targeting CLEC12A, has been explored to overcome T-cell exhaustion and improve efficacy in AML [94] (NCT06128044). Disruption of other inhibitory molecules has been investigated mainly in solid tumors, with potential extrapolation to AML [95,96].

Similarly, knockout of Regnase-1 (ZC3H12A), a regulator of cholesterol metabolism and immune signaling, reprograms T cells into long-lived effector cells with enhanced persistence, metabolic fitness, and anti-tumor function. This strategy has demonstrated efficacy in both solid and hematological malignancies [97,98]. Additionally, disruption of cytokine-inducible SH2-containing protein (CISH), a negative regulator of cytokine signaling, has shown promise in NK cell-based CAR therapies, including CAR-NK cells targeting CLEC12A [99].

### 8.2. Disruption to Enhance Safety

Unlike strategies focused on modifying CAR T cells directly, gene editing of hematopoietic stem and progenitor cells (HSPCs) offers an alternative route to improve safety in AML therapies (Figure 5b). This approach mitigates on-target, off-tumor toxicity by removing the expression of CAR target antigens from healthy cells. For example, CRISPR/Cas9-mediated deletion of CD33 in HSPCs preserved hematopoietic function while rendering cells resistant to CD33-directed CAR T cell cytotoxicity, enabling leukemia clearance without prolonged myelosuppression [100,101,102]. Similarly, epitope editing of CD45 via base editing has rendered HSPCs resistant to CD45-targeted CAR T cells while maintaining CD45 expression and normal hematopoiesis, paving the way for universal CAR T cell therapies against hematologic malignancies [103]. Although these modifications are not intrinsic to CAR T cells themselves, they represent a promising complementary strategy to expand the therapeutic window of AML-directed CAR therapies.

### 8.3. Disruption to Prevent Fratricide

Fratricide, caused by CAR T cells targeting antigens they themselves express, can hamper manufacturing and therapeutic efficacy. To address this, gene editing is used to disrupt the target antigen within the CAR T cells (Figure 5c). Knockout of CD45, CD97, or CD7 via CRISPR/Cas9 has prevented fratricide, enabling robust CAR T cell expansion and activity in both preclinical and clinical models [104,105,106] (NCT04538599).

### 8.4. Disruption to Achieve Allogeneic Compatibility

Autologous CAR T cell therapies require individualized manufacturing that is costly, time-consuming, and sometimes unfeasible in heavily pretreated patients or in AML subsets where immune cells serving as the starting material are often compromised. Allogeneic (“off-the-shelf”) CAR T cells from healthy donors overcome these limitations. Gene editing is pivotal (Figure 5d): T cell receptor alpha chain constant/T cell receptor beta chain constant (TRAC/TRBC) disruption eliminates TCR expression to prevent alloreactivity, β_2_ microglobulin (B2M) knockout reduces host rejection, and CD52 disruption confers resistance to alemtuzumab-based lymphodepletion. Together, these edits enhance persistence and function of allogeneic CAR T cells [4,91,92,93]. Clinical applications are ongoing, including CAR T cells targeting CD123 [107,108,109] (NCT03203369, NCT03190278), CLEC12A (NCT06128044), CD33 [60] (NCT05942599) and potentially FLT3 [110].

### 8.5. Overexpression of Key Components

Overexpression of transcriptional and signaling regulators represents another gene-modification strategy (Figure 5e). Enforced expression of PGC1α, a master regulator of mitochondrial biogenesis, restores oxidative capacity in metabolically exhausted T cells, improving persistence and cytotoxicity [111]. Similarly, BATF overexpression, in cooperation with IRF4, counteracts exhaustion and promotes the formation of long-lived memory CAR T cells [112]. In AML, c-JUN overexpression has rescued defective antigen receptor signaling and enhance antileukemic activity; notably, this approach is currently being evaluated in a phase I clinical trial of CD33-directed CAR T cells [113] (NCT04835519).

## 9. CAR-Non-T Cell

While gene editing continues to refine CAR T cell design, fundamental challenges limit T-cell-based therapies in AML. Antigen heterogeneity, the hostile bone marrow microenvironment, and risks of on-target, off-tumor toxicity underscore the need for complementary immune platforms. To overcome these barriers, the field has expanded beyond conventional αβ T cells, harnessing alternative effectors such as NK cells, macrophages, γδ T cells, MAIT cells, and NKT cells for CAR engineering. These “CAR-non-T cell” approaches aim to combine the versatility of CAR design with the unique biology of diverse immune subsets, potentially broadening the therapeutic landscape in AML.

### 9.1. CAR-NK Cells

NK cells are innate cytotoxic lymphocytes capable of rapid responses against malignant or infected cells. Unlike T cells, NK cells do not require MHC-mediated antigen recognition but instead integrate activating and inhibitory receptor signals. Their low risk of graft-versus-host disease (GvHD) makes them an attractive “off-the-shelf” platform. Moreover, CAR-NK cells have been associated with lower incidences of CRS and ICANS than CAR T cells, suggesting a safer profile in AML [114,115].

Preclinical studies have evaluated CAR-NK cells targeting CD33 [116], CD123 [117,118], TIM-3 [119], and FLT3 [120]. Next-generation designs incorporate checkpoint disruption, such as KLRC1 (NKG2A) knockout in CD33-directed CAR-NK cells [121] or combined CBLB/NKG2A/TIGIT disruption in CD276-specific CAR-NK-92 cells [122]. Combination strategies, including dual CD33/CD70 CAR-NK cells with proteasome inhibitors, have shown enhanced antileukemic activity [123]. Notably, CAR-NK cells recognizing neoepitopes from NPM1 mutations in HLA-A2^+^ AML cells and CAR-NK cells targeting mismatched MHC molecules like HLA-DR [124] highlight novel directions.

This evidence has driven multiple clinical trials (NCT04623944, NCT02742727, NCT02944162, NCT05215015, NCT05601466, NCT05665075), reflecting the active translation of CAR-NK strategies into the clinic.

### 9.2. Other CAR-Non-T-Cell

Beyond NK cells, several immune subsets are under evaluation [115]:CAR-macrophages (CAR-M): Highly phagocytic and antigen-presenting, macrophages infiltrate hypoxic niches and remodel the microenvironment via cytokine secretion and T-cell priming. CAR-Ms have shown efficacy in solid tumors [114,125], though AML applications are limited.CAR-γδ T cells: γδ T cells combine innate cytotoxicity with adaptive features, recognizing stress ligands and phosphoantigens via non-polymorphic MHC-related molecules instead of classical MHC. mSCF-CAR and CD123-CAR γδ T cells have shown antileukemic activity in preclinical models [126,127].CAR-MAIT cells: Semi-invariant T cells restricted by MR1, enriched in mucosal and hepatic tissues. Their cytokine responsiveness and low GvHD risk support interest as CAR carriers, though AML applications remain unexplored [128].CAR-NKT cells: Invariant NKT cells share features of NK and T cells, recognizing CD1d-presented lipids. Advances in HSPC-derived CAR-NKT generation have enabled scalable products [129]. In AML models, allogeneic CD33-directed CAR-NKT cells demonstrated efficient marrow homing and cytotoxicity against CD33-high and CD33-low blasts [130].

Collectively, alternative immune platforms expand cellular immunotherapy beyond αβ T cells, offering new solutions to overcome the barriers limiting CAR T cell efficacy in AML. Innate immune platforms such as CAR-NK cells may be particularly advantageous in those AML subsets where severe baseline T cell exhaustion compromises the functionality and manufacturability of autologous CAR T products.

## 10. Conclusions and New Horizons

Since their inception, CAR-based therapies have transformed the therapeutic landscape for malignancies once considered incurable, achieving unprecedented success in B-lineage cancers. However, their application to AML remains limited by the absence of leukemia-specific antigens, antigenic overlap with HSPCs, and the profoundly immunosuppressive nature of the bone marrow microenvironment.

Continuous efforts are being made to expand the repertoire of targetable AML antigens. Ideal AML targets should meet several criteria, including high and homogeneous expression on leukemic blasts and leukemic stem cells; minimal or absent expression on healthy HSPCs and vital non-hematopoietic tissues; stable surface accessibility and membrane localization; and limited propensity for antigen downregulation or shedding. Beyond conventional myeloid markers such as CD33, CD123, and CLEC12A, numerous novel candidates have emerged, including CD117, CD19, CD276, CD38, CD44v6, CD7, CD70, FLT3, folate receptor β, IL1RAP, Lewis Y, NKG2D ligands, Siglec-6, TIM-3, and WT1, among others. Although many of these antigens have been investigated in both preclinical and clinical settings, none has yet demonstrated sufficient efficacy and safety to warrant broad clinical implementation. Given the intrinsic antigenic overlap between AML and healthy hematopoietic tissues, it is increasingly unlikely that a single, leukemia-specific target will emerge; instead, future advances are more likely to arise from combinatorial antigen strategies, context-dependent or neoantigen-derived targets, and engineering platforms capable of safely exploiting the “imperfect antigens” that AML biology affords.

Consequently, a wave of innovative CAR engineering strategies has emerged. Multi-antigen approaches aim to mitigate antigen escape and enhance selectivity; gene-editing technologies allow precise disruption/overexpression of key genes to improve efficacy and safety; regulatable CAR formats incorporate ON/OFF switches or suicide systems to enable real-time control of activity; and alternative immune cell platforms such as NK, NKT, or γδ T cells are being explored to bypass αβ T-cell-related limitations and reduce the risk of GvHD. Together, these advances are redefining the armament of cellular immunotherapy for AML, attempting to bridge the gap between promising preclinical advances and future clinical application. Moreover, these emerging CAR engineering strategies may ultimately allow tailored approaches for the highest-risk AML genotypes where standard treatments are ineffective.

Regarding CAR T cell trials in AML, most remain early-phase studies conducted in heavily pretreated R/R cohorts and are primarily designed as a bridge to allogeneic hematopoietic stem cell transplantation (allo-HSCT). This reflects both the limited durability observed with single-antigen CAR constructs and the frequent, sometimes prolonged, myelosuppression following infusion [6,7,61]. As next-generation platforms continue to advance, this paradigm may shift, with the goal of achieving more effective antileukemic activity without obligate transplantation and of refining patient selection toward those most likely to benefit.

However, the rapid expansion of next-generation cellular platforms has created a landscape of theoretically promising options, but without a clear framework to guide when and how each strategy should be deployed. Although antigen heterogeneity, selectivity requirements, and T-cell fitness offer some translational clues, a coherent rationale to prioritize among these technologies remains lacking, highlighting the need for structured early-phase evaluation to determine which platforms are best suited for specific biological and clinical contexts.

Evaluating these approaches at lower disease burden, such as in minimal residual disease (MRD)-positive remission, may improve CAR T cell expansion, persistence, and overall therapeutic performance [2,6]. In addition, deploying these enhanced platforms in genomically defined high-risk AML subsets could further increase the potential for clinical benefit and support CAR-based strategies beyond the current bridge-to-allo-HSCT paradigm [7,61]. Several trials implementing these next-generation platforms are currently ongoing, although mature efficacy and safety data remain limited (Appendix A).

Despite this progress, the translation of next-generation cellular platforms remains constrained by substantial manufacturing and scalability barriers. Autologous CAR T cell production in AML is frequently hindered by poor T cell fitness in heavily pretreated patients, leading to manufacturing delays or even production failures [7]. Allogeneic approaches introduce additional layers of complexity, including multiplex gene editing to prevent GvHD and host rejection, optimized expansion and cryopreservation workflows, and the need for Good Manufacturing Practice (GMP)-compliant quality controls that are not yet standardized across centers [31,33]. Similarly, many of the most biologically compelling technologies, such as logic-gated circuits, armored or modulable constructs, specific gene-editing strategies, and non-T-cell CAR platforms, require sophisticated engineering and manufacturing processes that are considerably more intricate than those supporting approved CD19- or BCMA-directed CAR T cell products.

Although considerable limitations remain, the rapid evolution of CAR technologies, coupled with deeper knowledge into AML biology and the bone marrow niche, holds the promise of achieving durable remissions and potentially curative outcomes. Continued collaborative efforts linking synthetic biology, fundamental and clinical immunology, and precision medicine will be essential to fully realize the transformative potential of CAR-based therapies in AML.

## Figures and Tables

**Figure 1 cancers-17-03892-f001:**
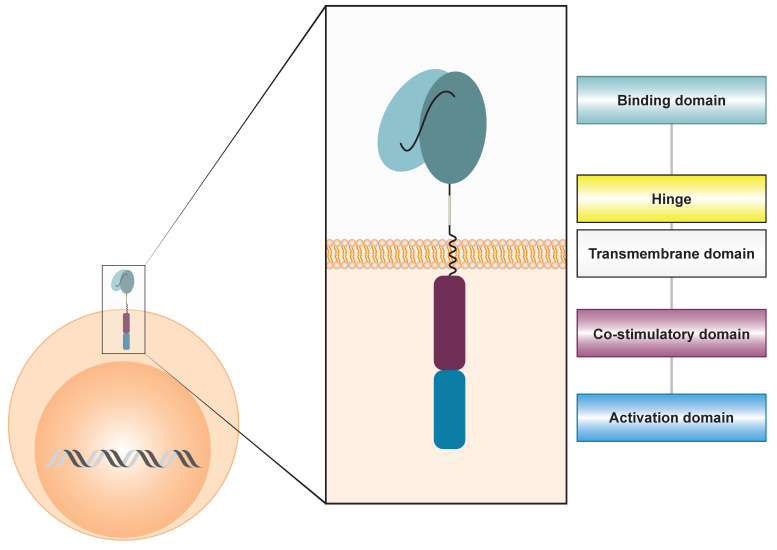
Canonical CAR architecture. Representation of a second-generation CAR expressed on the lymphocyte membrane, composed of a scFv derived from a monoclonal antibody, connected via a hinge and a transmembrane domain to a co-stimulatory signaling domain and the intracellular CD3ζ signaling domain.

**Figure 2 cancers-17-03892-f002:**
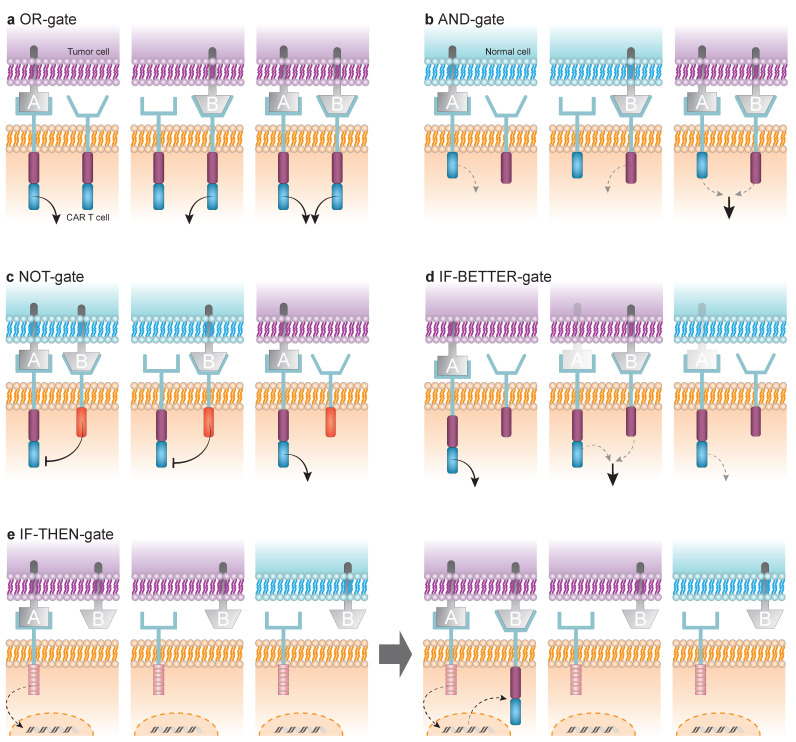
Logic-gated CAR T cells. (**a**) OR gate: coexpression of fully functional CARs targeting tumor antigens A and B. (**b**) AND gate: coexpression of partially functional CARs; full T-cell activation occurs only upon simultaneous engagement with antigens A and B coexpressed on tumor cells, but not on normal cells. (**c**) NOT gate: coexpression of a fully functional CAR specific for antigen A and an inhibitory CAR (iCAR) specific for antigen B. T-cell activation occurs when the CAR engages antigen A expressed exclusively on tumor cells, whereas iCAR engagement with antigen B on normal cells reversibly suppresses CAR T cell activity. (**d**) IF-BETTER gate: coexpression of a fully functional CAR specific for antigen A and a chimeric costimulatory receptor (CCR) specific for antigen B. Full activation is triggered when CAR binds antigen A at high density on tumor cells; if antigen A is expressed at low density (more transparent antigen A), CCR engagement with antigen B on the same tumor cells is required for full activation. (**e**) IF-THEN gate: coexpression of a SynNotch receptor specific for antigen A. SynNotch engagement with antigen A (**left**) induces transient expression of a fully functional CAR specific for antigen B (**right**).

**Figure 3 cancers-17-03892-f003:**
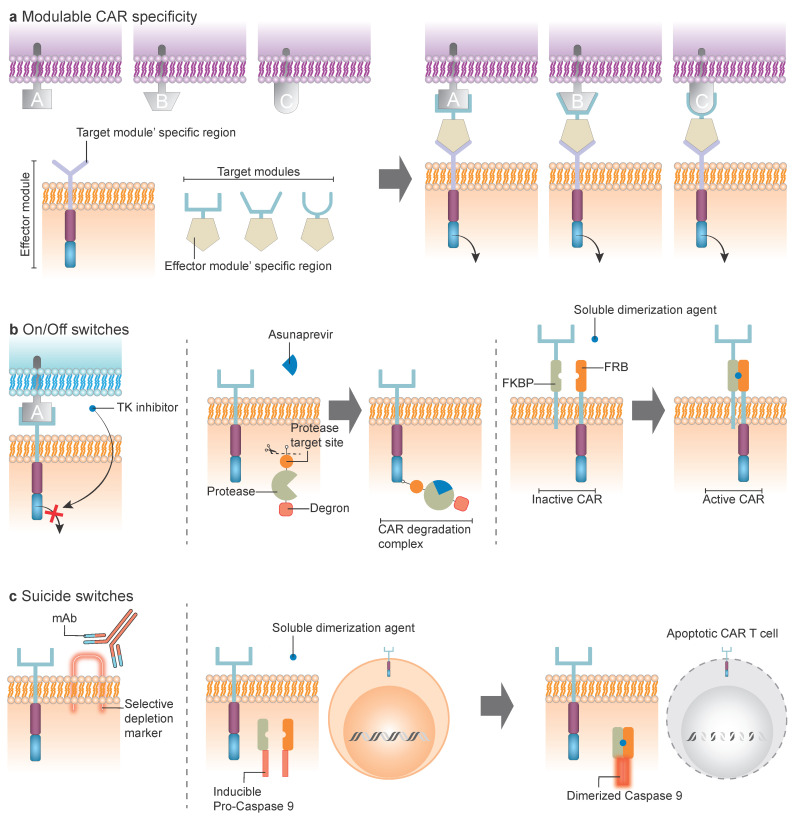
Modulable CAR platforms. (**a**) Modulable CAR specificity: CAR specificity can be externally controlled through two-component systems composed of a membrane-bound CAR-like receptor and a soluble adapter molecule that recognizes a tumor antigen. Both components are engineered to exhibit mutual affinity. Signaling occurs only when the adapter engages both the target antigen and the CAR. Antigen specificity for antigens A, B, or C can be modulated by administering the corresponding adapter molecule. (**b**) On/off switches: CAR activity can be reversibly modulated through tyrosine kinase (TK) inhibitors (**left**), CAR degradation complex regulated by a protease/protease inhibitor pair, allowing reversible control of CAR surface expression using asunaprevir (**middle**); and drug-induced dimerization of split CAR designs, in which separate CAR domains assemble into a functional CAR receptor only in the presence of a small molecule (**right**). (**c**) Suicide switches: Safety mechanisms enabling selective CAR T cell elimination by coexpression of depletion markers recognizable by monoclonal antibody-based therapies, and incorporation of nonfunctional inducible procaspase molecules that can be activated by a soluble dimerization agent, triggering caspase activation and CAR T cell apoptosis.

**Figure 4 cancers-17-03892-f004:**
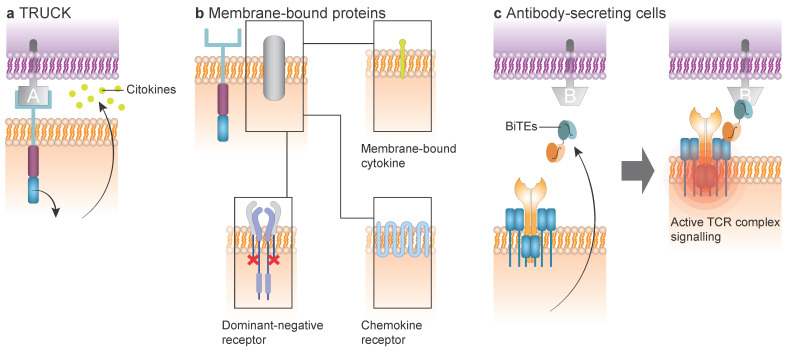
Armored CAR approaches. (**a**) TRUCKs: CAR T cells engineered to release specific cytokines that enhance CAR function and modulate the tumor microenvironment. (**b**) Membrane-bound proteins: CAR activity can be augmented by expression of membrane-bound cytokines, chemokine receptors, or dominant-negative inhibitory receptors that does not mediate inhibition. (**c**) Antibody-secreting CAR T cells: CAR T cells can be engineered to secrete antibody-based molecules, such as bispecific T-cell engagers (BiTEs), thereby recruiting bystander T cells against tumor antigens.

**Figure 5 cancers-17-03892-f005:**
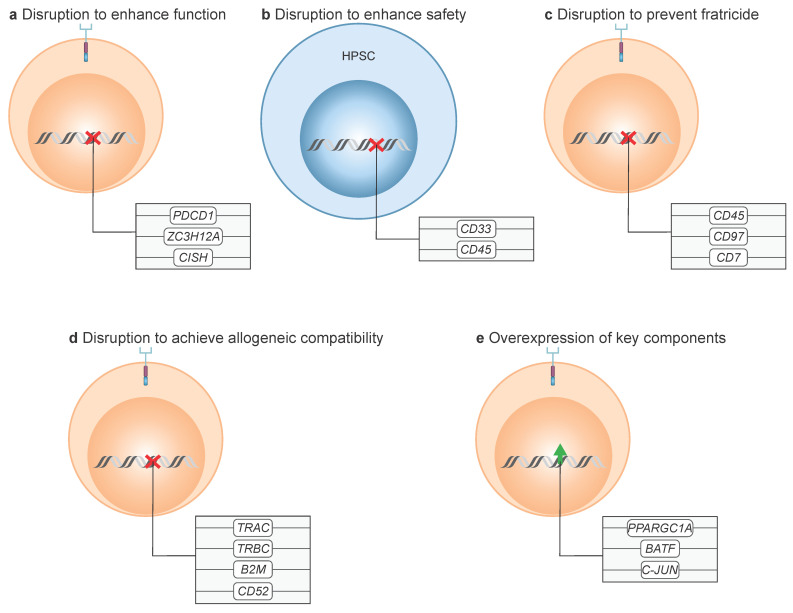
Gene-modified CAR T cells. (**a**) Disruption to enhance function: Targeted disruption of genes such as PDCD1, ZC3H12A, or CISH improves CAR T cell activity. (**b**) Disruption to enhance safety: Gene editing of hematopoietic stem and progenitor cells (HSPCs), such as deletion of CD33 or epitope editing of CD45, protects HSPCs from CAR T cell-mediated toxicity. (**c**) Disruption to prevent fratricide: Knockout of genes normally expressed on the T cell surface prevents CAR T cell fratricide. (**d**) Disruption to achieve allogeneic compatibility: Editing of histocompatibility-related genes enables the generation of allogeneic CAR T cell therapies. (**e**) Overexpression of key components: Overexpression of transcriptional regulators such as PPARGC1A, BATF, or C-JUN enhances CAR T cell functionality.

## Data Availability

No new data were created or analyzed in this study. Data sharing is not applicable to this article.

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
