# Peer review of "The Rise of Fine-Tuned CAR-Based Therapies Against Acute Myeloid Leukemia"

_cancers, 2025, doi:10.3390/cancers17243892_

Round 1
Reviewer 1 Report
Comments and Suggestions for Authors
Segura-Tudela and colleagues present a review manuscript on cellular therapy approaches in AML. The review is reasonably well written and I enjoyed reading the review. Cellular therapy in AML remains still in its nascency, unlike in B-lymphoid malignancies, however it appears to be the most sensible frontier to explore. Significant efficacy results are still lacking though several early phase trials including several cellular therapy products and approaches are ongoing in myeloid malignancies. Given the antigenic heterogeneity, oligoclonal biology and immunosuppressive microenvironment in myeloid malignancies such approaches have not garnered promise thus far.
Few comments:
- While the authors nicely describe the different approaches with CAR T and NK cells, a context on their usability in AML could be a prudent opening paragraph. This could be AML genotypes where they are expected to gain traction (TP53 mutated, MECOM rearranged, R/R AML, post MPN AML) where there is a dire lack of targeted treatment options and dismal outcomes. Do the authors feel cellular therapy approaches can fare better in some genomic subtypes of AML? AML with TP53 aberrations have significant T-cell exhaustion at baseline and autologous CART might not be potent in these situations. Please consider discussing.
- Can the authors also discuss how such cellular therapy approaches can be adopted in better trial design and in the future in actual AML therapy: which are the settings where they should be explored frontline; should such approach be used as definitive consolidation or a bridge to allogeneic HSCT (the latter seems to be the case presently for most situations). This will have some overlap with Point 1.
- It would benefit summarizing data from some cellular therapy trials in AML that have published results (mostly as abstracts as we are aware). One paragraph should suffice.
- What are the challenges the authors think would be with scalability of cellular therapy approaches in AML? Some of the approaches the authors have mentioned are biologically exciting but not as straight forward to scale compared to approved CD19/BCMA directed CAR T therapies.
Minor:
1. Page 4, Line 118: Change "Scape" to "Escape"
Author Response
Please see the attached response document.

Reviewer 2 Report
Comments and Suggestions for Authors
The manuscript by Segura-Tudela et al. (“The Rise of Fine-Tuned CAR-Based Therapies against Acute Myeloid Leukemia”) reviews the field of ex vivo engineered cell therapy for AML, focusing largely on the various approaches that have been tried preclinically and clinically. It provides an impressive breadth of coverage of the field and is well-organized and well written.
General comments/suggestions
This is more a comment: As the authors point out, success of CAR-Ts and other immunotherapies in AML has been elusive, in contrast to parallel efforts in other blood cancers. The authors present a dizzying array of options to improve the performance of basic CAR-T designs, but many have been tried and it is not clear what directions will be most fruitful. The authors don’t offer much guidance in that respect. Is this simply a case where clinical trials will need to be continually prosecuted until something works? Are there translational cues that could help guide and prune the number of possible combinations of options that they describe? If they do not have opinions, perhaps it is worth pointing out the challenges of simply having so many options that could be tested in the clinic.
Specific comments/suggestions
- There are a few typos, but the main diction issue I have pertains to the use of the word “allogenicity.” I think it means the propensity to induce allogeneic reaction, but is used in this review in the opposite way.
- I would recommend softening the conclusions of the section titled “Modulation of CAR affinity,” perhaps by pointing out that these studies do not control all the variables and the numbers are driven mostly by CLDN18.2 and GD2. Thus, the conclusions may not be so generalizable.
- In the section on OR gates, it is not clear if in the CAR-T trials conducted to date in AML instances of antigen escape have been observed. If so, what does it mean? Are OR gates the solution to a problem that already exists, or a theoretical one? Or has some other issue (e.g., lack of deep responses) hindered the emergence of resistant AML clones?
- It is my understanding that γδ T cells not not recognize “stress ligands and phosphoantigens independently of MHC.” Rather, they are displayed on non-polymorphic MHC paralogs.
- The authors state that “continuous efforts are being made to expand the repertoire of targetable AML antigens.” Do they really believe that after all this effort over the past decades that better antigens remain to be discovered? This is an example where the authors might take a stand.
Author Response

(The authors gave the same response as above.)

Reviewer 3 Report
Comments and Suggestions for Authors
This manuscript presents a timely and valuable review on the development status of CAR-T cell therapy for refractory Acute Myeloid Leukemia (AML). The systematic approach of compiling and discussing various studies, particularly by categorizing them based on CAR modification methods, is helpful for the reader's understanding.
However, while the review covers a vast amount of ongoing global research, it currently lacks depth and the authors' own critical perspective.
Major Comments
Lack of Depth and Critical Analysis: The paper provides a comprehensive overview of various research efforts. However, the authors largely list these studies without delving into the specific details, mechanisms, or clinical challenges of each approach. Consequently, the content feels somewhat superficial.
Insufficient Discussion on Future Directions and Author's Perspective: The paper would be significantly improved by incorporating the authors' insights on which approaches are most promising, what the hurdles to clinical translation are, and which strategies have a high likelihood of practical application. More emphasis on the authors' critical analysis and vision for the future of AML CAR-T therapy is needed to elevate the paper's impact and originality. The current originality of the review appears limited.
Minor Comments
Literature Search and Selection Methodology: Please provide a more detailed description of the methodology used to select the papers included in this review (i.e., inclusion/exclusion criteria, search terms, and databases). This would enhance the transparency and rigor of the literature search.
Formatting Inconsistency: Please check and correct the font size in lines 318–319, as it appears to be larger than the surrounding text.
Round 2
Reviewer 1 Report
Comments and Suggestions for Authors
I thank the authors for addressing my comments adequately. I have no further comments
Reviewer 3 Report
Comments and Suggestions for Authors
Thank you for providing the revised manuscript and your detailed responses to my previous comments.
I appreciate that you have adequately addressed the majority of my critiques. Specifically:
You have repeatedly and effectively emphasized the various challenges facing CAR-T cell therapy for AML and the immature nature of the current clinical data in the conclusion.
The description of the methods used for paper selection has been added, which promotes clarity.
The requested adjustments to the font size have been appropriately implemented.
The addition of the Supplemental Table S1 is useful and enhances the reader's understanding.
I am satisfied with the revisions made in response to my feedback. No further revisions are required from the authors.